# Cytomegalovirus Specific Serological and Molecular Markers in a Series of Pregnant Women with Cytomegalovirus Non Primary Infection

**DOI:** 10.3390/v14112425

**Published:** 2022-10-31

**Authors:** Claire Périllaud-Dubois, Emmanuelle Letamendia, Elise Bouthry, Rana Rafek, Isabelle Thouard, Corinne Vieux-Combe, Olivier Picone, Anne-Gaël Cordier, Christelle Vauloup-Fellous

**Affiliations:** 1Université Paris Cité and Université Sorbonne Paris Nord, Inserm, IAME, F-75018 Paris, France; 2Virology Laboratory, AP-HP.Sorbonne Université, Hôpital Saint-Antoine, F-75012 Paris, France; 3Department of Neonatal Medicine, Hôpital Antoine Béclère, APHP Université Paris Saclay, DMU2 Santé des Femmes et des Nouveau-nés, F-92140 Clamart, France; 4Virology Laboratory, CHU Angers, F-49000 Angers, France; 5Virology Laboratory, AP–HP.Université Paris-Saclay, Hôpital Paul-Brousse, F-94804 Villejuif, France; 6Division of Obstetrics and Gynecology, AP-HP.Nord, Hôpital Louis Mourier, F-92700 Colombes, France; 7Department of Obstetrics and Gynecology, Tenon Hospital, AP-HP.Sorbonne Université, F-75020 Paris, France; 8Université Paris-Saclay, INSERM U1193, 94804 Villejuif, France

**Keywords:** CMV non primary infection, pregnant women, serology, CMV PCR, serum, cCMV

## Abstract

(1) Background: In a period where systematic screening of CMV during pregnancy is still debated, diagnosis of non primary infection (NPI) remains challenging and an obstacle to systematic screening. Our aim is to report kinetics of serological and molecular CMV markers of NPI. (2) Methods: We identified immunocompetent pregnant women with CMV NPI as women known to be seropositive for CMV before pregnancy who gave birth to cCMV infected infants. We performed CMV-IgG, CMV-IgM, CMV-IgG avidity and CMV PCR retrospectively on sequential serum samples collected during pregnancy. (3) Results: We collected 195 serum samples from 53 pregnant women with NPI during pregnancy. For 29/53 (55%) patients, no markers of active infection were observed (stable IgG titers, negative IgM and negative PCR). CMV PCR was positive in at least one serum for 18/53 (34%) patients and median viral load was 46 copies/mL, IQR (21–65). (4) Conclusions: For more than half of patients with confirmed CMV NPI during pregnancy, available diagnostic tools are liable to fail in detecting an active infection. These should therefore not be used and universal neonatal screening for CMV remains the only way to detect all cCMV infections.

## 1. Introduction

Cytomegalovirus (CMV) is the most frequent worldwide cause of congenital viral infection with a prevalence estimated between 0.5 and 1% of all live births. Congenital CMV (cCMV) is a major cause of sensorineural hearing loss and mental retardation [1,2,3]. Vertical CMV transmission from mother to fetus can occur after primary (PI) or non-primary (NPI) maternal infection. After PI, the transmission rate from mothers to fetuses is estimated around 40% but varies depending on gestational age at maternal CMV infection [3,4,5]. After NPI, the transmission rate to the fetus remains unknown, as maternal diagnosis of NPI is not routinely performed. However, it is now established that risk of sequelae among cCMV children is unrelated to the type of maternal infection [6,7,8]. At birth, 13% of cCMV neonates are symptomatic mainly with growth restriction, microcephaly, ventriculomegaly, chorioretinitis, sensorineural hearing loss, hepatitis, thrombocytopenia and a purpuric skin eruption [2,9]. Additionally, approximately 15 to 20% of all cCMV children will develop long-term disability, mainly late-onset hearing loss and other developmental disorders. In countries where seroprevalence is around 50% such as France, half cCMV cases are due to maternal PI and half to maternal NPI. But in countries with high maternal seroprevalence, almost all cCMV are due to NPI [10,11,12,13].

Diagnosis of CMV PI during pregnancy mainly relies on serology: detection of specific CMV-IgG and IgM, associated with CMV-IgG avidity in the case of positive CMV-IgM [14]. Diagnosis of CMV NPI is usually performed in the follow-up of immunocompromised patients at risk of severe CMV NPI, and relies on CMVPCR in whole blood or plasma [15,16]. Previous studies already highlighted difficulties in detecting NPI in immunocompetent pregnant women with currently available serological tools, and diagnosis of NPI in this context remains challenging [17,18].

Since 2016, a universal neonatal screening of cCMV is offered to all newborns in our III-level maternity [13]. This has allowed us to retrospectively identify those born to mothers with positive CMV-IgG prior to pregnancy. Our aim in this article is to report kinetics of serological and molecular CMV markers in 53 immunocompetent women with NPI during pregnancy.

## 2. Materials and Methods

### 2.1. Population

Patients retrospectively included in our study were pregnant women with the two following criteria: (1) documented maternal CMV immunity before pregnancy and (2) confirmed cCMV infected fetus/neonate. In our III-level maternity, between 2016 and 2021, 41 cCMV infants, born from 40 pregnant women with CMV NPI during pregnancy, were born infected and were identified by neonatal systematic screening for cCMV [13]. Moreover, we identified 13 other pregnant women with CMV NPI during pregnancy in another French maternity, because of fetal or neonatal cCMV diagnosis. Neonatal diagnosis of cCMV infection was performed with a positive CMV PCR in urine samples.

Serum samples collected during pregnancy for these 53 patients were retrospectively analyzed between June 2016 and September 2021 in our virological laboratory (Hôpital Paul Brousse). Serum samples were previously stored at −40 °C until analysis.

### 2.2. Serology

CMV-IgG and CMV-IgM and CMV-IgG avidity were measured in our laboratory with LIAISON XL (LXL, DiaSorin^®^, Saluggia, Italy) and/or VIDAS (bioMérieux^®^, Craponne, France) following the manufacturers’ recommendations. High CMV-IgG avidity measured on a serum sample collected before 10 weeks of gestation confirmed immunization prior to pregnancy. We defined a significant rise of CMV-IgG as a doubling of antibody titer. Consequently, CMV-IgG were considered stable if titers did not double.

### 2.3. CMV DNA Detection and Quantification

CMV PCR was performed from 200 µL of serum. For nucleic acid extraction, the automated system QIASymphony (Qiagen, Germantown, MD, USA) with the DSP virus/pathogen minikit was used following manufacturer recommendations. For amplification and detection, the Rotor-Gene Q (Qiagen) with Artus^®^ CMV QS-RGQ kit (Qiagen) was used, following manufacturer recommendations.

## 3. Results

### 3.1. Results Description

#### 3.1.1. Number of Serum Samples per Patient

We collected 195 serum samples from 53 pregnant women with NPI during pregnancy. Median number of serum per patient was 4 with 1, 1 and 2 samples for respectively the first, second and third trimester of pregnancy (Appendix A).

#### 3.1.2. CMV Serological and Molecular Profiles

Before 10 weeks of gestation, all patients had positive CMV-IgG and high CMV-IgG avidity index. We performed CMV-IgG, CMV-IgM, CMV-IgG avidity and CMV PCR on all samples available. We distinguished four panels according to the number of potential markers of NPI: Panel 1 (no marker); Panel 2 (1 marker); Panel 3 (2 markers); Panel 4 (3 markers). Results are distributed as follows (Table 1, Table 2 and Table 3):

Panel 1: 29/53 (54.7%) women had stable CMV-IgG titers, negative CMV-IgM and negative CMV PCR in all samples; 

Panel 2: 3/53 (5.7%) women had significant rise in CMV-IgG titers but negative CMV-IgM and negative PCR in all samples;

Panel 3: 14/53 (26.4%) women had stable CMV-IgG titers but positive or equivocal CMV-IgM and/or positive CMV PCR in at least one serum sample;

Panel 4: 7/53 (13.2%) women had significant rise in CMV-IgG titers, positive CMV-IgM and positive CMV PCR in at least one serum sample.

Overall, 29/53 (54.7%) patients had no marker of active CMV infection in any serum sample all along pregnancy (panel 1), and 24/53 (45.3%) had at least one marker of infection in at least one serum sample collected during pregnancy (panels 2, 3, 4). Remarkably, 53/96 (55.2%) samples collected from these 24 women were negative for both CMV-IgM and CMV PCR. CMV PCR was positive in at least one serum sample for 18/53 (34.0%) patients and median viral load was 46 copies/mL, IQR (21–68). Positive CMV-IgM were detected in 7/53 (13.2%) women and a significant rise in CMV-IgG in 10/53 (18.9%) women.

## 4. Discussion

Our study focuses on CMV NPI during pregnancy in 53 women that gave birth to a cCMV infected child. In most studies, diagnosis of recurrent infection is based on a significant rise in CMV-IgG (with high CMV-IgG avidity) and/or the presence of CMV-IgM although these are also observed in other clinical situations, and more frequently than in CMV NPI cases: non-specific stimulation of the immune system, maternal auto-immune disorders, and other cross-reacting herpetic infections [19,20]. In our study, positive CMV-IgM were detected in only 7/53 (13.2%) women (7/21 with positive IgM among the panels 3 + 4, Table 1) and a significant rise in CMV-IgG in only 10/53 (18.9%) cases (Table 1, panels 2 + 4). Moreover our results highlight that most patients with NPI (54.7%) had stable CMV-IgG titers, negative CMV-IgM and negative PCR. Therefore, currently available diagnostic tools most often fail in detecting NPI in immunocompetent pregnant women.

For 18/53 (34%) women, PCR was positive in at least one sample. Viremia in NPI during pregnancy, as in immunocompetent non-pregnant individuals, seems to be transient, and very low viral loads were detected (median: 46 copies/mL, IQR [21–68]) [21,22]. One limitation of our study is that we performed CMV PCR on serum samples, which is less sensitive than whole blood samples; however, blood samples are not routinely collected from pregnant women. It is therefore likely that we underestimated PCR positivity rate. Our team recently highlighted performances and limits of CMV PCR in serum of immunocompetent patients [23]. By contrast to CMV positive-IgM, when CMV PCR is positive, whatever the viral load, NPI is confirmed. However, when CMV PCR is negative, this cannot exclude NPI. Indeed, NPI may be a reactivation, that is, a reappearance of an endogenous CMV strain acquired before pregnancy. In this situation, reactivation in the macrophages of the uterus [24], in the cervix, and in the kidneys is certainly responsible for most fetal infection, and such local reactivation is unlikely to be linked with maternal viremia [25]. Reinfection, on the other hand, is an infection with a new viral strain and is thought to lead to maternal viremia. To differentiate reactivation from reinfection is difficult and up till now, is based on the appearance of new antibody specificity against polymorphic epitopes of CMV [26,27].

Maternal NPI are responsible for half of cCMV infections in France [10] and it is now established that the higher the CMV seroprevalence is, the higher is the observed prevalence for cCMV [28]. Diagnosis of NPI during pregnancy thus represents a real challenge. Herein lies the strength of our study: for 53 patients with NPI during pregnancy, we investigated kinetics of serological and virological markers throughout pregnancy, and these markers were not conclusive for NPI diagnosis. Studies on this particularly topic are scarce. Picone et al. had similar results in a previous study on 9 cases [18], and Hadar et al. [29] found positive CMV-IgM in 3/12 (25%) women with NPI, compared to 14% in our study. Many clinicians suspect NPI in the case of positive CMV-IgM or in the case of significantly raised IgG titers (doubling of titer). Additionally, most will be reassured if CMV-IgM is negative or CMV-IgG stable. Our results, however, indicate that neither CMV-IgG kinetics nor CMV-IgM can allow the confirmation or exclusion of NPI. A positive CMV PCR (34% positivity rate in our study) certainly indicates an active infection and in our series would have confirmed NPI, yet it is clearly not sensitive enough. All clinicians and biologists should be aware that currently available serological and virological tools are unreliable for diagnosing NPI during pregnancy. Next steps would be to develop/identify reliable markers to detect NPI, in order to discuss treatment with valaciclovir as it is now offered to pregnant women with PI [30,31].

Overall, given that CMV NPI during pregnancy cannot be detected with our standard tools and that NPI represent at least half maternal infections leading to cCMV infections, it is essential to perform PCR in amniotic fluid in the case of ultrasound findings (such as cerebral findings, in utero growth retardation, etc.) even where there is no indication of recent PI in a CMV-immune woman [32]. Finally, these observations encourage the consideration of universal neonatal screening for cCMV in order to detect all cCMV infections and offer to these children the appropriate follow-up [13].

## Figures and Tables

**Table 1 viruses-14-02425-t001:** Distribution of serological and molecular profiles of the 53 pregnant women with CMV non-primary infection (195 serum samples, distributed according to the panel: none, one, two or three markers).

	Panel 1	Panel 2	Panel 3	Panel 4
Details of Markers	Stable IgG Titer Negative IgM Negative PCR	Rising IgG Titer Negative IgM Negative PCR	Stable IgG Titer A Least 1 Serum with: Positive IgM and/or Positive PCR	Rising IgG Titer A Least 1 Serum with: Positive IgM and/or Positive PCR
n patients	29/53	3/53	14/53	7/53
% patients	54.7%	5.7%	26.4%	13.2%
n serums/total serums (%)	99/195 (50.8%)	12/195 (6.2%)	58/195 (29.7%)	26/195 (13.3%)

**Table 2 viruses-14-02425-t002:** Details of serological and molecular markers of active infection in the 58 serum samples of panel 3 (2 markers).

Details of Markers	Stable IgG Titer Negative IgM Negative PCR	Stable IgG Titer Positive IgM Negative PCR	Stable IgG Titer Negative IgM Positive PCR	Stable IgG Titer Positive IgM Positive PCR
n serums (%)	24/58 (41%)	16/58 (28%)	14/58 (24%)	4/58 (7%)

58 serums from 14 patients that had stable IgG titer and at least one serum during pregnancy with positive IgM and/or positive PCR (2 markers). Among these 58 serums, 24/58 (41%) had no marker, 30/58 (52%) had one marker and 4/58 (7%) had two markers.

**Table 3 viruses-14-02425-t003:** Details of serological and molecular markers of active infection in the 26 serum samples of panel 4 (3 markers).

Details of Markers	Rising IgG Titer Negative IgM Negative PCR	Rising IgG Titer Positive IgM Negative PCR	Rising IgG Titer Negative IgM Positive PCR	Rising IgG Titer Positive IgM Positive PCR
n serums (%)	17/26 (65%)	3/26 (12%)	5/26 (19%)	1/26 (4%)

26 serums from 7 patients that had rising IgG titer and at least one serum during pregnancy with positive IgM and/or positive PCR (3 markers). Among these 26 serums, 17/26 (65%) had one marker, 5/26 (31%) had two markers and 1/26 (4%) had 3 markers.

## Data Availability

Data can be provided by the authors on demand.

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
