# Peer review of "Cytomegalovirus Specific Serological and Molecular Markers in a Series of Pregnant Women with Cytomegalovirus Non Primary Infection"

_viruses, 2022, doi:10.3390/v14112425_

Round 1
Reviewer 1 Report
Perillaud-Dubois and co-authors present a clear and straightforward retrospective analysis of CMV detection in seropositive mothers with cCMV infants. This study is important as it uses a much larger cohort (53 patients) each with multiple serum samples (at minimum one per pregnancy trimester) which allows longitudinal analysis. My comments are focused on clarity of presentation and I see no issues with study design.
Main comments:
1) There is no text description for Table 2 or Table 3. Please describe (at least briefly) this data.
2) The data in the discussion is somewhat hard to track back to the data shown in the tables, it may help to reference the table and panel. For example, line 134: "7/53 (13.2%) of women [Table 1, panel 4] and a signifcant rise [in] CMV-IgG in only 10/53 (189%) [of] cases [Table 1, panel 2+panel 4]". As well as other relevant locations: lines 138, etc.
3) Did all pregnancies result in full-term births?
Minor comments:
1) This journal is a general virology journal. For clarity, it would be helpful to avoid the use of nonstandard acronyms (i.e. WG (line 96 only).
2) Line 27 states that 55% of patients are non-detectable for active CMV infection, while line 30 says that this is "most" patients. This is an overinflation of the results. 55% is more than half (and a significant finding in its own right) but definitely not most.
3) Sentence on line 118 is an overview of the data shown in Table 1 and may be better placed as an introduction to this description (i.e. moved before line 96 for clarity).
4) Line 174 introduces a suggestion of "perform PCR in amniotic fluid in case of ultrasound findings". This is confusing - why would this be indicated? What type of ultrasound findings? 1-2 additional sentences and/or references might clarify this.
Author Response
We thank the reviewer for his carefully reading of the manuscript.
Main comments:
1/ There is no text description for Table 2 or Table 3. Please describe (at least briefly) this data.
We added brief description for Table 2 and Table 3
Lines 129-131: “58 serums from 14 patients that had stable IgG titer and at least one serum during pregnancy with positive IgM and/or positive PCR (2 markers). Among these 58 serums, 24/58 (41%) had no marker, 30/58 (52%) had one marker and 4/58 (7%) had two markers.”
Lines 137-139: “26 serums from 7 patients that had rising IgG titer and at least one serum during pregnancy with positive IgM and/or positive PCR (3 markers). Among these 26 serums, 17/26 (65%) had one marker, 5/26 (31%) had two markers and 1/26 (4%) had 3 markers.”
2/ The data in the discussion is somewhat hard to track back to the data shown in the tables, it may help to reference the table and panel. For example, line 134: "7/53 (13.2%) of women [Table 1, panel 4] and a signifcant rise [in] CMV-IgG in only 10/53 (189%) [of] cases [Table 1, panel 2+panel 4]". As well as other relevant locations: lines 138, etc.
We added these details in the discussion:
Lines 148-150: “In our study, positive CMV-IgM were detected in only 7/53 (13.2%) women (7/21 with positive IgM among the panels 3+4, Table 1) and a significant rise in CMV-IgG in only 10/53 (18.9%) cases (Table 1, panels 2+4).”
Details for reviewer: Concerning the 7 women with positive IgM, they were not only those in panel 3. They were divided between panel 3 and panel 4, for a total of 7/53 women with positive IgM at least for 1 serum during pregnancy.
3/ Did all pregnancies result in full-term births?
This information is not available.
Minor comments:
1/ This journal is a general virology journal. For clarity, it would be helpful to avoid the use of nonstandard acronyms (i.e. WG (line 96 only).
We have written it down in full. Line 98
2/ Line 27 states that 55% of patients are non-detectable for active CMV infection, while line 30 says that this is "most" patients. This is an overinflation of the results. 55% is more than half (and a significant finding in its own right) but definitely not most.
We have nuanced our statement:
Line 30: “For more than half of patients”
3/ Sentence on line 118 is an overview of the data shown in Table 1 and may be better placed as an introduction to this description (i.e. moved before line 96 for clarity).
We placed this sentence Lines 100-102 as suggested.
Moreover, we added a description for Table 1: Lines 122-123:
“195 serum samples from 53 patients, distributed according to the panel (none, one, two or three markers).”
4/ Line 174 introduces a suggestion of "perform PCR in amniotic fluid in case of ultrasound findings". This is confusing - why would this be indicated? What type of ultrasound findings? 1-2 additional sentences and/or references might clarify this.
We added examples of ultrasound findings and added a reference (32). Indeed, as maternal NPI can not be diagnosed in pregnant women and as NPI can provide congenital CMV infection (and also symptomatic cCMV infections).
Therefore, it is indicated and common to perform amniocentesis in case of ultrasound abnormalities suggestive of cCMV in women with a history of positive CMV serology before pregnancy (including years before pregnancy).
Lines 190-192:
“it is essential to perfom PCR in amniotic fluid in case of ultrasound findings (such as cerebral findings, in utero growth retardation, …) even if no argument of recent PI in CMV-immune women [32] »
Reviewer 2 Report
Authors aimed to find kinetics of serological and molecular CMV markers of NPI and enrolled 53 pregnant women who gave birth to cCMV infants. Authors newly discovered 34% of patients tested positive for serum PCR tests at least once, but regretfully concluded that available diagnostic tools are liable to fail in detecting an active infection in NPI women. This study determined only frequencies of serological and molecular markers of cCMV in NPI
To find serological and molecular CMV markers of cCMV infants in NPI mothers, these have to be compared with the markers of non-cCMV infants in NPI mothers. The case-control study can be performed to find specific markers for cCMV infants in NPI mothers including odds ratios, 95% CI, and statistical significances.
Not only serological and molecular markers, but also clinical symptoms in mothers, fetal growth restriction, and abnormal ultrasound findings may be additional candidates for cCMV markers in NPI pregnant women.
Author Response
Response to reviewer:
The aim of our study was to identify wether our virological tools (serology and CMV PCR) were able to diagnose maternal NPI during pregnancy.
Indeed, some clinicians are regularly confused by positive CMV-IgM during pregnancy or by rising IgG titers and then suspect maternal NPI. We demonstrated here that these are not markers of NPI.
Moreover, we could ask wether CMV PCR could be performed prospectively to detect maternal NPI. Here again, we demonstrated that this is not sensitive enough.
Finally, we aimed breaking some dogmas and proving scientifically that they are wrong.
In this way, we identified congenitally infected neonates born from mother with preexisting immunity for CMV before pregnancy, so with maternal non primary infection (NPI) leading to congenital infection. We retrospectively analyzed her serum samples (results presented in the article).
We specified the way to diagnose congenital infection in material and methods section. Lines 73-74: “Neonatal diagnosis of cCMV infection was performed with a positive CMV PCR in urine samples.”
What suggests the reviewer (to compare markers of NPI in transmitters and non-transmitters women) is impossible, precisely because we can not detect maternal NPI during pregnancy.